# Quantitative Loop-Mediated Isothermal Amplification Detection of *Ustilaginoidea virens* Causing Rice False Smut

**DOI:** 10.3390/ijms241210388

**Published:** 2023-06-20

**Authors:** Yu Zhang, Xinyue Li, Shuya Zhang, Tianling Ma, Chengxin Mao, Chuanqing Zhang

**Affiliations:** Department of Plant Pathology, Zhejiang Agriculture and Forest University, Hangzhou 311300, China; zy1335659339@163.com (Y.Z.); lixinyue1992@126.com (X.L.); 15168349553@163.com (S.Z.); zafumaocx@163.com (C.M.)

**Keywords:** rice false smut, quantitative loop-mediated isothermal amplification (q-LAMP), detection, ustiloxins biosynthetic gene

## Abstract

Rice false smut caused by *Ustilaginoidea virens* is one of the most devastating diseases in rice worldwide, which results in serious reductions in rice quality and yield. As an airborne fungal disease, early diagnosis of rice false smut and monitoring its epidemics and distribution of its pathogens is particularly important to manage the infection. In this study, a quantitative loop-mediated isothermal amplification (q-LAMP) method for *U. virens* detection and quantification was developed. This method has higher sensitivity and efficiency compared to the quantitative real-time PCR (q-PCR) method. The species-specific primer that the UV-2 set used was designed based on the unique sequence of the *U. virens* ustiloxins biosynthetic gene (NCBI accession number: BR001221.1). The q-LAMP assay was able to detect a concentration of 6.4 spores/mL at an optimal reaction temperature of 63.4 °C within 60 min. Moreover, the q-LAMP assay could even achieve accurate quantitative detection when there were only nine spores on the tape. A linearized equation for the standard curve, y = −0.2866x + 13.829 (x is the amplification time, the spore number = 10^0.65y^), was established for the detection and quantification of *U. virens*. In field detection applications, this q-LAMP method is more accurate and sensitive than traditional observation methods. Collectively, this study has established a powerful and simple monitoring tool for *U. virens*, which provides valuable technical support for the forecast and management of rice false smut, and a theoretical basis for precise fungicide application.

## 1. Introduction

Rice false smut is a disease affecting rice spikes that occurs from the flowering to the milking stage [1,2]. Its most typical and visible symptom is the replacement of rice grains with false smut balls [3,4]. It occurs mainly in Asian countries such as China, Japan, Korea, the Philippines, and India, and is one of the most devastating diseases in the world’s major rice producing regions [5,6,7,8]. In recent years, due to the promotion of short-stalked compact and high-yielding rice varieties, indica–japonica interspecific hybrid rice combinations, changes in cultivation patterns, and the excessive use of nitrogen fertilizer during the tillering and gestation periods, the occurrence of rice false smut has become increasingly serious and has gradually risen from a previously minor or sporadic disease to become one of the three new major diseases affecting rice in China [9,10].

The damage caused not only results in a decrease in rice quality and yield, but also the generation of mycotoxin ustiloxins on infected rice spikelets [11,12]. As antimitotic cyclopeptide mycotoxins, the ustiloxins produced within a false smut ball can inhibit microtubule assembly and cell skeleton formation, which poses a serious threat to farmland preservation and ecosystems, as well as the health of humans and animals [13]. Strategies to manage this devastating disease are therefore urgently needed.

It is widely accepted that *Ustilaginoidea virens* (teleomorph *Villosiclava virens*) is the causal agent of rice false smut [14,15]. As a typical airborne disease, virulent pathogen spores land on the surface of a rice spikelet and germinate hyphae as well as false smut balls on the spikelet [3,12,16,17,18]. Thus, the epidemic of rice false smut is closely related to the amount of *U. virens* spores in the field, and the diagnosis of rice false smut, combined with accurate detection and spore quantification, is of great importance for its prevention and management [5,19]. Traditionally, the microscopic counting of spores after capture is widely used in rice false smut diagnosis; however, this method requires specialist taxonomic technicians [20]. Given the complexity of environmental samples and human subjectivity, it is difficult to obtain reliable data with high efficiency via microscopic analysis. A quantitative real-time PCR (q-PCR) technique has been applied for the early identification and quantification of pathogens in airborne diseases [21]. However, this technique is susceptible to interference from environmental dust and other pathogens, making it difficult to quantify the low concentrations of spores captured [20].

Loop-mediated isothermal amplification (LAMP), developed by Notomi et al. in 2000, is a non-PCR-based nucleic acid amplification technique that can be used for the molecular detection of various bacteria, viruses, fungi in disease diagnosis [22,23,24]. The LAMP reaction is carried out at a constant temperature (between 60 and 65 °C) in less than an hour through the use of two pairs of primers—an inner primer (FIP/BIP) and an outer primer (F3/B3). These two pairs of primers constitute the basic LAMP primer set for the LAMP reaction, in order to recognize specific nucleic acid sequences of monitored targets [25,26,27]. An additional pair of LAMP primers, loop primers, can also be used to significantly improve LAMP efficiency. A simple and visual LAMP assay was developed by Yang et al. in 2018 for the rapid diagnosis of *U. virens* [28]. However, this assay cannot be used directly for quantitative detection of complex DNA samples. The quantitative-LAMP (q-LAMP) assay (DiaSorin S.p.A., Saluggia, Italy) is a technical improvement from the classical LAMP, which combines LAMP technology with the real-time fluorescence quantitative PCR technique [29]. It is based on the addition of nucleic acid fluorescent dyes, such as SYBR Green or SYTO, resulting in a more sophisticated method suitable for the needs of field diagnosis [30,31]. In this study, we aimed to design and develop a specific and sensitive q-LAMP assay for detection and quantification *U. virens*, which can be applied in the early diagnosis of rice false smut for preventing the spread of this devastating airborne disease. Additionally, this study is the first report to describe a quantitative diagnostic test for the detection of *U. virens* using q-LAMP.

## 2. Results

### 2.1. Design of Primers for U. virens Detection

The best LAMP UV-2 primers were designed based on the ustiloxins biosynthetic gene sequence of *U. virens* (NCBI accession number: BR001221.1) that did not show any similarities to other sequences available in the National Center for Biotechnology Information (NCBI) GenBank database, in order to allow specific amplification of *U. virens* (Figure 1, Table 1) [32,33,34]. Additionally, the UV-2 primer sets met the requirement that ΔG values must be less than or equal to −4 Kcal/mol at the 3′end of F3/B3 and F2/B2 and 5′ends of F1c and B1c.

### 2.2. Optimization of the q-LAMP Assay

To optimize the q-LAMP assay system, the q-LAMP assay was carried out using the UV-2 primer sets at temperatures ranging from 61.8 °C to 66 °C. As shown in Figure 2, the fluorescence quantitative results show that the strongest fluorescence intensity and the shortest reaction time were obtained when the reaction temperature was 63.4 °C (which reached the amplification peak at 30 min). Thus, 63.4 °C was chosen as the reaction temperature at which to carry out the optimal q-LAMP assays.

### 2.3. Specificity Validation of the q-LAMP Assay System

The specificity validation of design in the q-LAMP assay system (using UV-2 primer sets and 63.4 °C as reaction temperature) was tested using the *U. virens* stain and the other nine fungi. The results show that fluorescence signals were detected in the samples with the DNA template of *U. virens*, while the samples with the DNA template of the other nine fungi or ddH_2_O (negative control) did not show any fluorescence signal (Figure 3), indicating that the design of the q-LAMP assay system was highly specific to the detection of *U. virens*.

### 2.4. Sensitivity Validation of the q-LAMP Assay System

The sensitivity validation of the q-LAMP assay was determined using the genomic DNA of gradient dilution of *U. virens* spores as templates under optimal conditions (using primer sets UV-2 and 63.4 °C as reaction temperature). As shown in Table 2 and Figure 4, the fluorescence signals were detected in the samples with 2 × 10^4^ spores/mL, 4 × 10^3^ spores/mL, 8 × 10^2^ spores/mL, 1.6 × 10^2^ spores/mL, 32 spores/mL, and 6.4 spores/mL within 60 min (the spore extracts were used as DNA template in q-LAMP assay system), while no signals were detected in the sample with the DNA template of 1.28 spores/mL. Thus, theoretically, the q-LAMP assay was able to detect the sample with a concentration of 6.4 spores/mL. We also compared the sensitivity of the q-LAMP assay system with quantitative real-time PCR (q-PCR) for *U. virens* detection. The q-PCR assay was carried out using the FB/B3 primer set and the effective amplification reactions were detected in samples with spore concentrations of 2 × 10^4^, 4 × 10^3^, 8 × 10^2^, 1.6 × 10^2^ spores/mL, but not 32 spores/mL (Appendix A). Thus, the q-LAMP assay system is more sensitive and efficient compared to the q-PCR system used in this study.

### 2.5. Establishment of a Standard Curve for q-LAMP Detection of U. virens

A standard curve between the amplification time (x) and the Log_10_ value of spore number (y) was constructed based on the q-LAMP assay: y = −0.2866x + 13.829 (Figure 5, Appendix A), the formula used for calculating spore number is 10^0.65y^, and the correlation coefficient R^2^ = 0.9942, showing a good linear relationship.

### 2.6. Application of q-LAMP Assay for U. virens Spore Calculation

The standard curve of q-LAMP was applied to calculate *U. virens* spore number on tapes, and each tape sample contained 450, 116, 29, and 9 manually added spores, respectively. As shown in Table 3 and Figure 6, the amplification times quantitated using the cycle threshold (Ct) values for the tested samples were 34.03, 37.12, 40.46, and 43.17, corresponding to 446.07, 118.51, 28.29, and 8.85 predicted spores per tape, respectively, which is very close to the actual spore number on each Melinex tape. Thus, this q-LAMP system can efficiently quantitate *U. virens* spore number with high accuracy.

### 2.7. Field Application of q-LAMP Assay System

The q-LAMP system results show that spores of *U. virens* were first observed on the 27 August 2018, while the results obtained using the microscope show that spores were observed for the first time on the 2nd of September. Then, the number of spores began to rise rapidly and reached its peak on the 20 September and obvious symptoms of rice false trot were found in the field on the 25th of September. In the following year (2019), the q-LAMP system and microscope manual observation were used to monitor the spores of *U. virens* in the field again. The results showed that the q-LAMP system detected the spores for the first time on the 31st of August, while microscopic observation led to the detection of only a handful of spores on the 6th of September, and the concentration of spores reached its peak on the 30th of September. The symptoms of rice false smut were found in the field on the 5th of October. Through monitoring the dynamic changes in the spore number of *U. virens* in the field for two consecutive years (Figure 7), it was clearly seen that the q-LAMP system was faster and more efficient than the traditional microscopic observation method.

## 3. Discussion

Currently, rice false smut disease caused by *U. virens* is one of the most devastating rice diseases in China, as well as many other countries [35]. The occurrence of rice false smut disease not only results in the decrease in rice quality and the serious loss of rice yield, but also threatens food safety due to its production of toxic mycotoxins within the false smut balls [10,11]. However, it has been found that rice false smut disease is difficult to control. As a typical airborne disease, the epidemic of rice false smut is closely related to the number of *U. virens* spores in the field; thus, early detection and warning are critical for preventing and mitigating rice false smut. In this study, a q-LAMP assay system was developed. The results show that the species-specific UV-2 primer sets in the q-LAMP assay system could correctly distinguish *U. virens* from the other nine air-dispersed fungi, including *Fusarium fujikuroi*, *F. oxysporum*, *F. proliferatum*, *F. solani*, *F. graminearum*, *Penicillium* sp., *Pyricularia oryzae*, *Alternaria alternata*, *and Rhizoctonia solani* (Figure 3). Additionally, sensitivity validation found that the q-LAMP assay was able to detect a concentration of 6.4 *U. virens* spores/mL at an optimal reaction temperature of 63.4 °C within 60 min (Figure 4), and the q-LAMP assay could even achieve accurate quantitative detection when there were only nine *U. virens* spores on the Melinex tape (Figure 6). Moreover, there was a good linear relationship between the spore amount (y) and the amplification time (x) (Figure 5), which enables accurate quantification of *U. virens* and early diagnosis of *U. virens* infection via q-LAMP assay.

The LAMP primer set consisted of two outer primers (forward primer F3 and backward primer B3), two inner primers (forward inner primer FIP and backward inner primer BIP), and two loop primers (forward loop F and backward loop B) (Appendix A). The outer primers (F3 and B3) were used in the initial steps of the LAMP reactions but later, during the isothermal cycling, only the inner primers were used for strand-displacement DNA synthesis. Outer and inner primers are necessary for LAMP primer design, while the loop primers can be used to accelerate amplification reactions and improve the LAMP assay efficiency [36]. In this study, the q-LAMP primer set was designed according to the work of Wang et al. [20] and Li et al. [37], containing a forward inner primer (FIP), a backward inner primer (BIP), and two outer (F3 and B3) primers. The *ustiloxins biosynthetic gene* sequence was used to design primers to ensure their specificity. Meanwhile, we sequenced the targeted region of *ustiloxins biosynthetic gene* in 15 *U. virens* stains and designed the primer sets elaborately to eliminate the interference from nucleotide polymorphisms, ensuring the amplification efficiency in *U. virens* detection (Figure 3).

For *U. virens* diagnosis, besides traditional disease diagnosis that includes the identification of symptoms, isolation of pathogens, and microscopic techniques, a conventional nested-PCR assay has been developed for the detection *U. virens* in rice [6]. However, the nested-PCR has less sensitivity and cannot be used in accurate quantification of *U. virens* [38]. Recently, the q-PCR technique and q-LAMP assay have been applied for the identification and quantification of pathogens in disease diagnosis. In this study, we have established these two systems for *U. virens* quantification. The q-PCR assay was carried out using the F3/FB primer set and the effective amplification reactions were detected in samples with spore concentrations of 2 × 10^4^, 4 × 10^3^, 8 × 10^2^, 1.6 × 10^2^ spores/mL, but not 32 spores/mL (Appendix A), indicating a lower sensitivity of q-PCR for *U. virens* detection compared to the q-LAMP assay system.

Rice false smut has no symptoms in the early stage and can only be identified in the late stage when the smut balls appear. Chemical control is the main means of rice false smut prevention and control [39]. The previous study showed that the first 4~15 d of ear bud breakage was the main period of control, and the first 4~7 d of control was the best [40]. If the key window in the infection of *U. virens* in rice is not grasped, the efficacy of management will be inadequate [16,40]. Therefore, for rice false smut that relies on airborne transmission, early detection and early warning can aid in disease prevention and control. In this study, we collected spore samples of *U. virens* from the field for two consecutive years using the q-LAMP assay system and microscopic observation. Compared with manual observation, the q-LAMP assay system could detect spores in the air more accurately and quickly, providing a theoretical basis for precise fungicide application (Figure 7). Therefore, the q-LAMP assay, with higher efficiency and sensitivity, is a better choice for the early diagnosis of rice false smut.

In conclusion, this is the first assay developed for the detection of *U. virens* using q-LAMP assays. Compared with other *U. virens* detection methods, the newly developed LAMP assay has superior operability, specificity, and sensitivity, and is more suitable for the quantitative detection of *U. virens* and early diagnosis.

## 4. Materials and Methods

### 4.1. Fungal Isolates

Isolates of *Ustilaginoidea virens* and the nine other fungal pathogens used in this study were isolated and identified in our lab, and detailed information on each fungus is listed in Table 4. Isolates were maintained on potato dextrose agar (PDA, prepared by 200 g potato, 20 g glucose, and 20 g agar per 1 L pure water) slants at 4 °C.

### 4.2. DNA Template Preparation from Mycelium and Spores for q-PCR and q-LAMP Analysis

Preparation of mycelial DNA template for optimum conditions and specificity of the q-LAMP assay, after mycelia grew covering two-thirds of the PDA plate surfaces, the hyphae were then transferred to a mortar and ground with liquid nitrogen. The resultant powder was then placed into a 2-mL centrifuge tube and the mycelial DNA of each fungus was extracted using a Genomic DNA Kit (Sangon Biotech Co., Ltd., Shanghai, China) according to the manufacturer’s instructions. The extracted DNA was used as DNA template in q-LAMP analyses and stored at −20 °C. For spore DNA template preparation, after growing on PDA medium at 25 °C in darkness for 20 days, 5 mm diameter mycelial plugs taken from colony margin were placed into the potato sucrose (PS, prepared by 200 g potato and 20 g sucrose per 1 L pure water) medium at 25 °C 150 rpm, in darkness for 7 days. Spores were separated from medium with filtration through four layers of lens tissue and washed twice with distilled water. Then, spores were diluted with 10% sodium dodecylsulfate (SDS) solution into a series of concentration gradients. An amount of 1-mL spore suspension sample of known concentration mixed with 200-μL 10% Chelex-100 solution [20], 50-μL 10% SDS solution and 0.4 g acid-washed glass beads was placed into a 2.0-mL centrifuge tube. The sample was lysed by Fast Prep Apparatus (JXFSTPRP-24L, Jingxin Technology, Shanghai, China) for 40 s at speed of 6 m/s and placed in boiling water bath for 5 min. The grinding and heating steps were repeated three times, after which the sample was placed on ice for 2 min. The cooled lysate was used directly as DNA template in q-PCR and q-LAMP analyses and stored at −20 °C.

### 4.3. Design of q-LAMP Primers for U. virens Detection

Ustiloxin A and Ustiloxin B of *U. virens* are synthesized by ustiloxins biosynthetic gene that was found to be species-specific to *U. virens* [13,41]. Thus, the sequence of ustiloxins biosynthetic gene (NCBI accession number: BR001221.1) was chosen for q-LAMP primer design using Primer Explore V5 (online web service, http://primerexplorer.jp/e/) ensuring the specificity and accuracy of q-LAMP assay system for *U. virens* detection. The q-LAMP primers contain forward inner primer (FIP), backward inner primer (BIP), and two outer (F3 and B3) primers (Appendix A). The primers were designed according to the following rules: ΔG values of less than or equal to −4 Kcal/mol at the 3′end of F3/B3 and F2/B2 and 5′ends of F1c and B1c.

### 4.4. Determination of Optimum Condition of the q-LAMP Assay

To better facilitate the efficiency of q-LAMP reaction, the LAMP reaction system was improved via screening for the optimal reaction temperature based on a reference from Notomi [42]. The LAMP reaction was carried out in the following reaction mixtures containing 0.25 μM·L^−1^ of the primers, FIP and BIP; 0.2 μM·L^−1^ of the primers, F3 and B3; 1.0 mM·L^−1^ betaine; 2.0 mM·L^−1^ dNTPs (Takara Bio Inc., 108, San Jose, CA, USA); 25 mM·L^−1^ Tris-HCl (pH 8.8); 12.5 mM·L^−1^ KCl, 12.5 mM·L^−1^ (NH_4_)_2_SO_4_; 10 mM·L^−1^ MgCl_2_; 0.125% (*v*/*v*) Triton X-100; 0.2 U·L^−1^ of Bst DNA polymerase (New England Biolabs, 110, Beijing, China); 0.5 μL 1 × SYBR Green I; and 1 μL of DNA template extracted as described above, and the volume was adjusted to 25 μL with nucleic-acid-free water. The screened reaction temperature gradients were 61.8 °C, 62.1 °C, 62.6 °C, 63.4 °C, 64.4 °C, 65.2 °C, 65.6 °C, and 66 °C. LAMP reactions were performed using a Bio-Rad quantitative fluorescent PCR instrument (Bio-Rad CFX96, Hercules, CA, USA) for 80 cycles each, each cycle for 60 s, and the reaction was terminated at 80 °C for 10 min. Optimal reaction temperature screening experiments were repeated three times.

### 4.5. Validation of the Specificity for q-LAMP Assay Systems

The specificity of the reaction system was tested by performing q-LAMP reactions at the optimal reaction temperature with UV-2 primers in above 25-μL reaction mixtures for 70 min. The assay results were compared with the DNA of *U. virens* and the 9 other fungi listed in Table 4. The nucleic acid-free water was set as negative control. Additionally, the DNA template of *U. virens* and the 9 other fungi were prepared as descripted in 4.2 mycelial DNA template preparation. The extracted DNA of *U. virens* and the 9 other fungi were stored at −20 °C and their concentration were more than 150 μg·mL^−1^. The results were rigorously validated with the assessment that the detectable peak of fluorescence signals detected by Bio-Rad CFX96 as positive; no fluorescence signal as negative. The specificity testing experiment was repeated three times.

### 4.6. Sensitivity Validation of q-LAMP and q-PCR Assay Systems

The sensitivity validation of q-LAMP reactions was performed at the optimal reaction temperature with UV-2 primers in reaction mixtures above 25-μL for 60 min. An amount of 1 μL of DNA lysate from *U. virens* spores of known concentration was used as a DNA template in the LAMP reaction system. The nucleic-acid-free water was used as a DNA template in the negative control (CK). The detectable peak of fluorescence signals detected by Bio-Rad CFX96 was regarded as positive, while no fluorescence signal was regarded as negative. Sensitivity assay experiments were repeated three times. The sensitivity of the q-PCR reaction system was assayed via performing q-PCR amplification using primers, UV-2 F3/B3. The q-PCR reaction system was 12.5 μL SYBR^®^ Premix Ex Taq II (Tli RNaseH Plus, 2×), 1.0 μL of forward primer F3 (10 μM), 1.0 μL of reverse primer B3 (10.0 μM), 1.0 μL DNA template (in CK, nucleic-acid-free water was used as DNA template), and the volume was adjusted to 25 μL with nucleic-acid-free water. The reaction conditions were: pre-denaturation at 95 °C for 2 min, denaturation at 95 °C for 5 s, annealing at 60 °C for 30 s, extension at 72 °C for 6 s. The fluorescence signal was collected during the extension for a total of 40 cycles, and finally the amplification curve was plotted. The detectable peak of fluorescence signals detected by Bio-Rad CFX96 was regarded as positive, while no fluorescence signal was regarded as negative. Sensitivity assay experiments were repeated three times.

### 4.7. Establishment of Standard Curves for q-LAMP Assay Systems

A standard curve was constructed using software SPSS 13.0 by analyzing the association of logarithmic values of the spore number (y) and the amplification time quantitated using the cycle threshold (Ct) values (x). The correlation coefficient R^2^ was used for assessing the linear relationship between the spore number in sample (y) and amplification time (x). The experiments were repeated three times.

### 4.8. Calculating of U. virens Spore Using q-LAMP System

Spores of *U. virens* were artificially added to each of the four Melinex tape (1 cm × 2 cm) in the ultra-clean bench, with 450, 116, 29 and 9 spores in each tape. The collected spore-adsorbed Melinex tape was cut and placed in 2-mL centrifuge tubes, and the genomic DNA of the spores on the Melinex tape was then extracted according to the method mentioned above. An amount of 1 μL of the cooled lysate was used directly as DNA template. q-LAMP assay was performed with the optimal reaction conditions in reaction mixtures above 25-μL for 60 min, and the time quantitated using the cycle threshold (Ct) values detected by Bio-Rad CFX96 was recorded as the amplification time (x). The linearized equation for the standard curve was used for converting the amplification time to the corresponding spore number. Then, the calculated spore number was compared to the amount of actual added (listed above) to test the accuracy efficiency of this q-LAMP system.

### 4.9. Field Application of q-LAMP Assay by U. virens

An air borne spore catcher (DIANJIANG, DJ-0723) with Melinex tape was established in Yongyou 1540 cultivation area, Jiangtang Village, Jinhua City, Zhejiang Province for the collection of spores of *U. virens*, and samples were collected at six-day intervals for 11 consecutive times, starting on the 9th of August 2018. Similarly, starting on the 13th of August of the following year (2019), 11 consecutive samples were collected every six days. The spores of *U. virens* were adsorbed on the Melinex tape and the tapes (1 cm × 1 cm) with spores were cut and placed in a 2-mL centrifuge tube, and the conidial DNA was extracted according to the methods mentioned above. An amount of 1 μL of the cooled lysate was used directly as DNA template. q-LAMP assay was performed with the optimal reaction conditions in the reaction mixtures above 25-μL for 60 min, and the amplification time quantitated using the cycle threshold (Ct) values was recorded. According to the established standard curve, the number of spores was calculated. The spore population of *U. virens* in the Melinex tape was recorded via q-LAMP assay at six-day intervals. Meanwhile, the spores of *U. virens* (1 cm × 1 cm) adsorbed on the slide were suspended in 1 mL of ddH_2_O, and the spore suspension was counted using a hemocytometer under the microscope to determine the spore concentration. There were three spore catchers placed at the collection site, and the data collected by each instrument were used as a repetition.

## Figures and Tables

**Figure 1 ijms-24-10388-f001:**
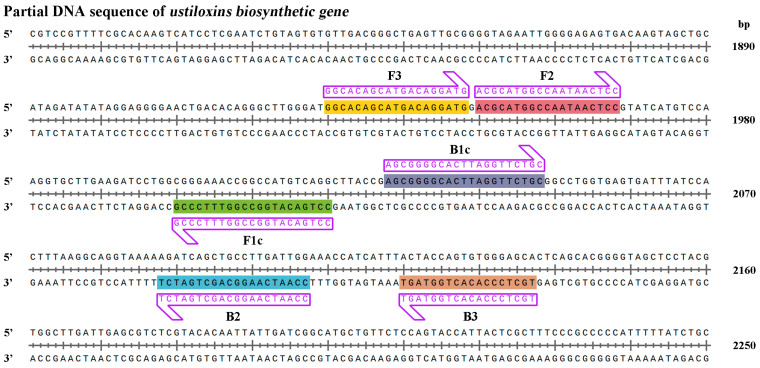
The species-specific primers for detecting *Ustilaginoidea virens* in the quantitative loop-mediated isothermal amplification (q-LAMP) and quantitative real-time PCR (q-PCR). The species-specific primers designed based on the sequence of the ustiloxins biosynthetic gene segments for identification and quantification of *U. virens* in q-LAMP assay and q-PCR assay. The forward and reverse primer sequences were highlighted with shade and arrow for orientation.

**Figure 2 ijms-24-10388-f002:**
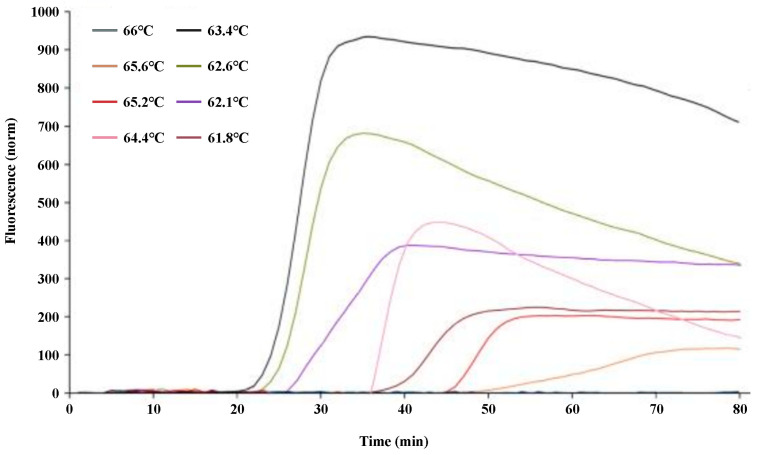
Optimization of the q-LAMP assay via reaction temperature screening. The influence of temperature ranged from 61.8 °C to 66 °C in the q-LAMP detection system and showed that the strongest fluorescence intensity and the shortest reaction time were obtained at 63.4 °C (black line).

**Figure 3 ijms-24-10388-f003:**
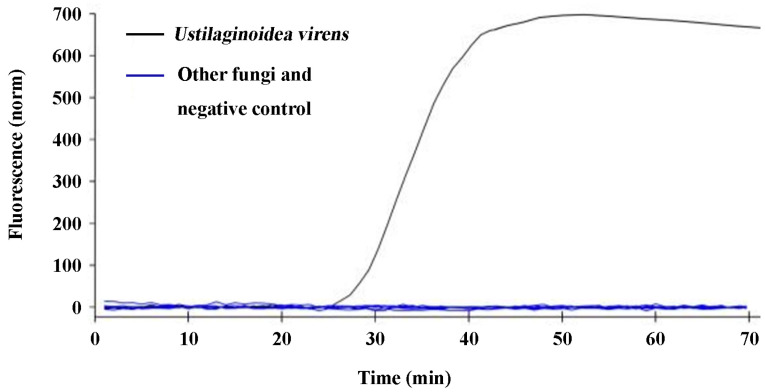
Specificity validation of the q-LAMP assay system. The q-LAMP assay system (using UV-2 primer sets and 63.4 °C as reaction temperature) was highly specific for the detection of *U. virens*. The q-LAMP assay system showed that fluorescence signals were only detected in the samples with DNA template of *U. virens* (black line), while the samples with DNA template of the other 9 fungi (including *Fusarium fujikuroi*, *F. oxysporum*, *F. proliferatum*, *F. solani*, *F. graminearum*, *Penicillium* sp., *Pyricularia oryzae*, *Alternaria alternata* and *Rhizoctonia solani*) or negative control (nucleic acid-free water) did not show any fluorescence signal.

**Figure 4 ijms-24-10388-f004:**
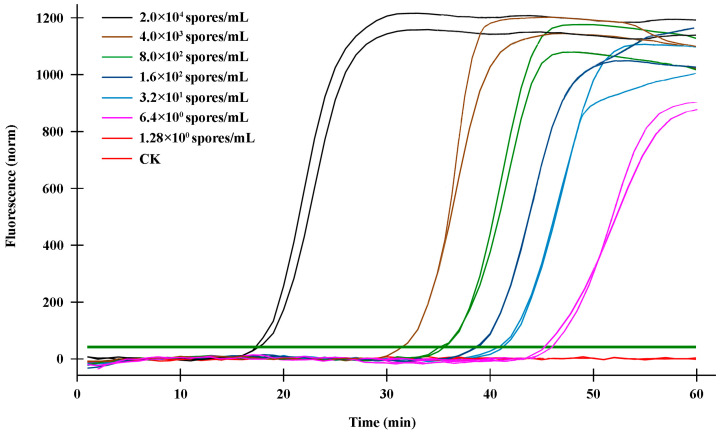
Sensitivity validation of q-LAMP system. The fluorescence signals in q-LAMP assays were detected in the samples with DNA template of 2 × 10^4^ spores/mL, 4 × 10^3^ spores/mL, 8 × 10^2^ spores/mL, 1.6 × 10^2^ spores/mL, 32 spores/mL, and 6.4 spores/mL within 60 min, while no signals were detected in sample with DNA template of 1.28 spores/mL and CK. The bolded dark green line (horizontal) indicates fluorescence threshold. Fluorescence signals above this threshold marked as a successful detection of *U. virens* in q-LAMP assays.

**Figure 5 ijms-24-10388-f005:**
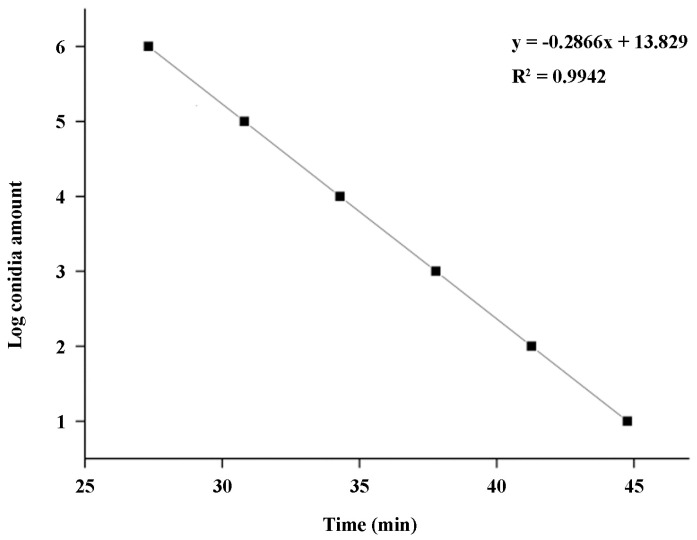
Standard curve of q-LAMP detection system. A standard curve between logarithmic values of the spore number (y) and the amplification time quantitated using the cycle threshold (Ct) values (x): y = −0.2866x + 13.829. The correlation coefficient (R^2^) is 0.9942, showing a good linear relationship.

**Figure 6 ijms-24-10388-f006:**
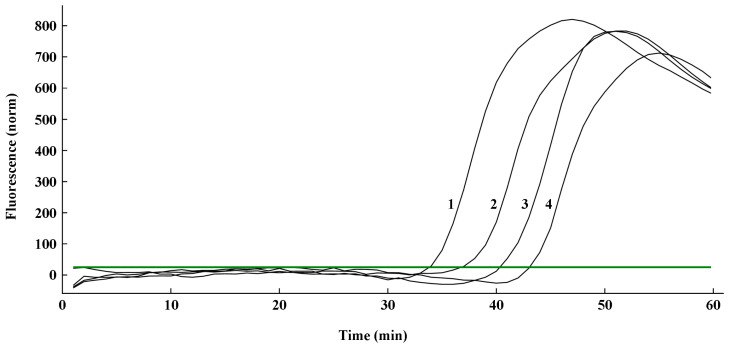
Quantitative detection of *U. virens* spores on Melinex tape using q-LAMP system. Serial numbers 1, 2, 3, and 4 represent 450, 116, 29, and 9 spores, respectively. The green line (horizontal) indicates fluorescence threshold. Fluorescence signals above this threshold marked as a successful detection of *U. virens* in q-LAMP assays.

**Figure 7 ijms-24-10388-f007:**
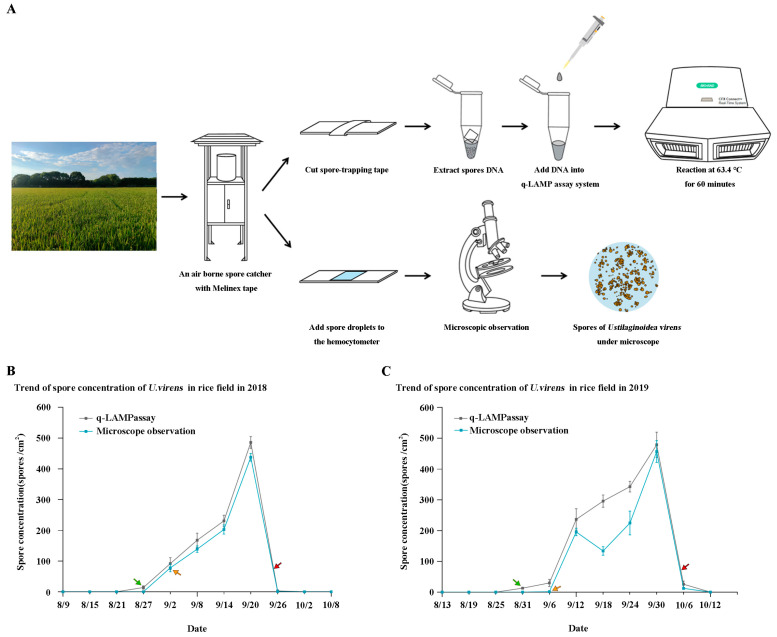
Field application of *U. virens* spores using q-LAMP system. (**A**) Flow chart of field *U. virens* spore sample detection, q-LAMP assay system and microscope observation were used for the collected samples, respectively. (**B**) The results of *U. virens* spore concentration measured by different methods in rice fields in 2018 (q-LAMP assay system is the gray line; microscope observation method is the blue line). (**C**) The results of *U. virens* spore concentration measured by different methods in rice fields in 2019 (q-LAMP assay system is the gray line; microscope observation method is the blue line). The green arrow is the first detection of spores by q-LAMP assay system, the orange arrow is the first observation of spores by microscope observation, and the red arrow is the occurrence time of rice false smut in the field.

**Table 1 ijms-24-10388-t001:** The sequences of species-specific primers used in the quantitative loop-mediated isothermal amplification (q-LAMP) assay and quantitative real-time PCR (q-PCR) assay.

Serial Number	Sequence (5′–3′)	
UV-2	F3	GGCACAGCATGACAGGATG
	B3	TGCTCCCACACTGGTAGT
	FIP(F1c-F2)	CCTGACATGGCCGGTTTCCCGACGCATGGCCAATAACTCC
	BIP(B1c-B2)	AGCGGGGCACTTAGGTTCTGCCAATCAAGGCAGCTGATCT

**Table 2 ijms-24-10388-t002:** The time for fluorescence signal to reach the fluorescence threshold and fluorescence signal records in the q-LAMP assays for testing samples with known spore concentration.

Spore Concentration (Spores/mL)	Time ^a^ (min) (Mean ± Standard Deviation)	Fluorescence Signals ^b^
2 × 10^4^	17.90 ± 1.64	+
4 × 10^3^	31.44 ± 0.71	+
8 × 10^2^	34.68 ± 1.26	+
1.6 × 10^2^	37.92 ± 1.53	+
3.2 × 10^1^	41.16 ± 0.98	+
6.4	44.40 ± 2.42	+
1.28		−
CK ^c^		−

^a^ Time for fluorescence signal reaching the fluorescence threshold in the q-LAMP assays. ^b^ “+” indicates a successful fluorescence signal detection, “−” indicates no fluorescence signal detected in the q-LAMP assays. ^c^ The nucleic acid-free water was used negative control (CK) in the q-LAMP assay.

**Table 3 ijms-24-10388-t003:** Quantitative detection of *U. virens* spores using q-LAMP system.

Ct ^a^	Manually Added Spores (Spores/mL)	Predictive Spores (Spores/mL)	R^2^	*p* Value
34.03	450	446.07	0.999	0.639
37.12	116	118.51		
40.46	29	28.29		
43.17	9	8.85		

^a^ the amplification times (x) quantitated using the cycle threshold (Ct) values.

**Table 4 ijms-24-10388-t004:** The information of strains used in the specificity validation of the q-LAMP assay system.

Species	Isolate NO.	Host	Origin
*Fusarium fujikuroi*	/	Rice	Zhejiang, China
*F. oxysporum*	ACCC30927 ^a^	Rice	Hainan, China
*F. proliferatum*	CICC2489 ^b^	Rice	Anhui, China
*F. solani*	ACCC37119	Rice	Hebei, China
*F. graminearum*	ACCC37680	Wheat	Jiangxi, China
*Penicillium* sp.	ACCC31507	Soil	Shandong, China
*Ustilaginoidea virens*	ACCC2711	Rice	Hunan, China
*Pyricularia oryzae*	ACCC37631	Rice	Fujian, China
*Alternaria alternata*	ACCC36843	Rice	Hainan, China
*Rhizoctonia solani*	ACCC36246	Rice	Beijing, China

^a^ ACCC (Agricultural Culture Collection of China). ^b^ CICC (China Center of Industrial Culture Collection).

## Data Availability

Not applicable.

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
