# Peer review of "Quantitative Loop-Mediated Isothermal Amplification Detection of Ustilaginoidea virens Causing Rice False Smut"

_ijms, 2023, doi:10.3390/ijms241210388_

Round 1

Reviewer 1 Report

The reviewed manuscript is dedicated to the design and validation of LAMP-based assay detecting Ustilaginoidea virens, a dangerous pathogen causing rice faultry smut and affecting amount and safety of harvested rice. Here, authors designed LAMP for quantitative detection of spores, assessed its sensitivity and specificity on a set of samples from various sites. The presented results are timely and interesting for scientists, specializing on the field of molecular diagnostics. However, a few comments need to be made and addressed.

 Major issues:

1.      The language of the manuscript needs to be polished in both terms of typos and style correction.

2.      Authors are encouraged to add more information in the Introduction about previously developed LAMP-based methods for detection of Ustilaginoidea virens (for example, 10.1111/jph.12991, 10.1094/PDIS-01-18-0065-RE).

3.      2.1. Design of primers for U. virens detection — it is known that additional pair of LAMP primers, loop primers, significantly increases the efficacy of LAMP (10.1006/mcpr.2002.0415). Were these primers used in the developed analysis? Also, nucleotide polymorphisms under primers could diminish the efficacy of amplification. Were the designed primers tested for any possible SNPs?

4.      2.2 Optimization of the q-LAMP assay — while SYBR Green I inhibits LAMP, several SYTO-dyes do not demonstrate such a negative effect (10.1016/j.mimet.2015.11.004, 10.2144/000114432). Usage of tolerable SYTO-dyes could increase sensitivity of the reported LAMP assay. Authors are also encouraged to provide results of gel-electrophoresis proving the length of accumulated LAMP products.

5.      Determination of optimum condition of the q-LAMP assay — CFX96 allows to perform melting curve analysis, which could help to distinguish specific and non-specific signals.

6.      Page 5, lines 133-134: “indicating the theoretically q-PCR assay was ineffective in detecting samples with spore concentration lower than 32 spores/mL.” — the qPCR assay was not optimized as it was done for qLAMP. Therefore, it would be better to change the statement and make it less strict.

7.      2.6. Application of q-LAMP assay for U. virens spore calculating — authors are encouraged to calculate a correlation coefficient between added and predicted numbers of spores.

8.      2.7. Field application of q-LAMP assay system — authors are encouraged to calculate a correlation coefficient between qLAMP and microscopy results.

Minor issues:

1.      Abstract: “Moreover, the q-LAMP assay can even achieve accurate quantitative detection when there were only 9 spores on the tape. A linearized equation for the standard curve, y = -0.2866x + 13.829 (x is the amplification time, the spore number = 100.65y), was established for detection and quantifying of U. virens.” — plausibly, it would be better to state limit of quantification and linear range of qLAMP without providing the equation in the Abstract. Mentioning of results obtained with samples from a rice field could also strengthen the abstract.

2.      Page 2, line 44: “telemorph” — teleomorph.

3.      Page 2, line 52: “Given that the complexity of environmental samples and human subjectivity” — plausibly, a verb is missing in this sentence.

4.      Page 2, line 60-61: “The LAMP reaction is carried out at a constant temperature (between 60 and 65°C) in less than an hour by use of two pairs of primers…” — currently, most LAMP-based methods apply 3 primers pairs.

5.      Page 2, line 67: “fluorescent probes (TaqMan probes…” —TaqMan probes (e.g. hydrolyzing probes) cannot be used for LAMP, as large fragment of Bst-polymerase has no 5’-3’ exonuclease activity.

6.      Figure 1 — the scheme of LAMP and qPCR primers at the B-panel could be omitted as it is a well-known information for specialists on the field.

7.      Plausibly, paragraphs 2.4 and 2.5 could be merged as they describe the results of the same experiment.

8.      Page 9, line 248: “Ustilaginoidea virens” — here and in several other places all species need to be italicized.

9.      Page 10, line 256: “DNA templet preparation” — template instead of templet.

10.  Page 10, line 265: “PS, prepared by 200 g potato and 20 g per 1 L pure water” — sucrose is missing.

11.  Page 10, line 266: “Spores separated from” — “were” is missing.

12.  Page 10, line 273: “heating septs were” — steps.

13.  Page 11, line 290: “0.25 μM·L-1 of the primers FIP and BIP” — commonly, FIP and BIP are added in LAMP in 1.6 or 0.8 μM concentrations. Lower concentrations of FIP and BIP primers decrease the efficacy of LAMP.

14.  4.5. Validation of the specificity for q-LAMP assay systems — DNA template concentration needs to be stated as it directly affects the results of the specificity assessment.

Please, find notions on the language of the manuscript in the section with general comments and suggestions.

Reviewer 2 Report

In this manuscript, the authors reported development of an effective diagnostic method to detect and quantify for spores of Ustilaginoidea virens using q-LAMP. It seems this method is very useful for early diagnosis for rice false smut in field. Authors designed experiments appropriately. So, I recommend to accept this MS for publishing in IJMS.

Author Response

Thanks for your comments,the revised contents in the manuscript was highlighted in red

Reviewer 3 Report

Review of the manuscript entitled “Quantitative Loop-mediated Isothermal Amplification Detec- tion of Ustilaginoidea virens Causing Rice False Smut” submitted to the MDPI, Molecular Microbiology. The authors had developed a quantitative LAMP approach for detection and quantification of U. virens spores. This method can be useful for diagnostics and treatment of the  rice false smut disease. In general the manuscript is well written, just a few grammar corrections may be needed. The Methods and Results sections need some edits. No template control is missing in one figure. No data shown for testing their new LAMP assay with mycelial DNA. It is unclear if LAMP reactions with the field collected spores were done with fresh or frozen samples.  It will be good to mention if any false positive reactions were observed. Strangely, I didn’t find a reference on paper describing LAMP detection of U. virens, see below.

The detailed comments:

L15: Consider primer set instead of sets.

L18: “achieve accurate quantitative detection when there were only 9 spores on the tape” How about greater numbers? You can mention this as well. 

L21: “faster than traditional observation method” You didn’t measure timing needed for the microscopic detection.

L71-73: “And this study is the also first report describing a quantitative diagnostic test for detection of U. virens using q-LAMP. Consider using “also the first…”

Figure 3: Use “other fungi” 

L118-119: Use Italics for the species names.

L123: Consider using “spore extracts”.

Table 2 , L136: It is confusing when you mention both, “The time for reaching fluorescence signal peak and fluorescence signal detection…” and have one Time column in the table. Which time is it?  I am not aware that BIO-RAD CFX96 software provides data for the time reaching fluorescence signal peak. Do you mean Cq? Do you show a mean time from several replicate reactions? Can you clarify this?  Also using “CK” for no template control and “/” for negative results is a little confusing.

Figure 5: It might be good to have the actual data points on the graph.

L170: “amplification peak” What do you mean? Cq?

Figure 6. No Template Control is missing. For LAMP  known to have false positive reactions it is very important to show NTC. Did you observe some cases of false positives?

Figure 7: Did you run q-PCR with these samples? Make 7B and 7C for these two graphs. Did you have replicate reactions for these samples? Show error bars then. 

L224-228: Why didn't you compare to the published qPCR methods (#37 from your references) but used a new one?

You didn’t mention the paper that also uses  LAMP detection of your pathogen.

 Yang, X., Al-Attala, M. N., Zhang, Y., Zhang, A. F., Zang, H. Y., Gu, C. Y., ... & Zhu, J. G. (2018). Rapid detection of Ustilaginoidea virens from rice using loop-mediated isothermal amplification assay. Plant disease, 102(9), 1741-1747 

I also suggest this paper for references: Zhou, Y. L., Izumitsu, K., Sonoda, R., Nakazaki, T., Tanaka, E., Tsuda, M., & Tanaka, C. (2003). PCR‐based specific detection of Ustilaginoidea virens and Ephelis japonica. Journal of Phytopathology, 151(9), 513-518.

L222: “ Recently” sounds not right for the 2013 paper (#37). You can add more recent references here. 

L248: Use Italics for the species name.

L257-261: No data shown that you tested mycelial DNA. You have to test the limit of detection with mycelial DNA and proof that it works. 

L274: What is the volume of the lysate?

Methods 4.3: I am just curious why you didn't design loop primers for more speedy reaction?

Methods 4.5: Did you validate the specificity of your qPCR reaction?

L323-324: The fluorescence signal was collected during the extension for a total of 40

cycles, and finally the lysis curve was plotted. Consider using signal was recorded. Why are you saying “lysis curve”?

Methods: Mention the number of replicate reactions.

Methods 4.9: How and for how long did you store the spore samples? Mention the number of replicate reactions. 

“Amplification time” Do you mean Cq?

“The spore population of U. virens in the glass slide was recorded by q-LAMP assay at six-day intervals.” It is confusing. What glass slide?

For Supplementary Fig S1 edit X-axis title, it should be “Cycles” for PCR graph.

Several places need improvement.

Round 2

Reviewer 1 Report

Many thanks to authors for their detailed reply on the comments and changes of the manuscript. Only a few comments still need to be made and addressed.

 Major issues:

1.      Authors are encouraged to add information about loop primers that they could increase the efficacy of the developed test. The same goes for information about absence of nucleotide polymorphisms under primers. This can be done in the Discussion section to provide more background for readers.

 Minor issues:

1.      Some minor typos still need to be corrected, like missed spaces before references, etc.

2.      “The forward inner primer (FIP), backward inner primer (BIP), and two outer (F3

2.and B3) primers constitute the basic LAMP primer set for LAMP reaction, and the

2.additional pair of LAMP primers, loop primers, significantly increases the efficacy of

2.LAMP. However, in this study, we have not developed the additional pair of LAMP

2.primers as the efficacy is efficient enough, however, this suggestion is a good point for

2.further improvements” — plausibly, it would be better to change the statement of question in the Introduction and mention that 2 or 3 pairs can be used for LAMP.

Please, find the comment about language in the Comments and Suggestions for Authors section.

Reviewer 3 Report

The revision of the author changes to the manuscript 

Thanks for improving the manuscript based on some of my comments. However, a few more serious issues (underlined) and minor issues still have to be addressed.

L 14: Check the grammar “primer sets UV-2 used was design”

L15-18: Your initial version was better than this sentence.

About your interpretation of the CFX96 data (Ct vs Cq). 

How do I know this machine produces these outputs for PCR reactions:

https://www.bio-rad.com/en-us/product/cfx96-touch-real-time-pcr-detection-system?ID=LJB1YU15

For LAMP reactions, the number of minutes (Time) instead of cycles is used. 

Do you have different outputs? 

I think you have to fix this issue (Ct vs Cq) throughout the manuscript. 

Fig 6 should have Time (min) on X-axis and no template control sample included.

Fig 5 and Supplementary table S1: I asked for the actual data points from 3 replicate reactions. but not the points on the regression line.

Fig 7. I asked about q-PCR tests with the field samples.

L 328: It should be an amplification curve.

Methods 4.9. Did you run LAMP reactions immediately after field spore collection?  Or do you store them for what time?

The results from the mycelium samples LAMP testing are not shown. You have to prove that this assay works for the mycelium samples as well.

Previously I mentioned: “For Supplementary Fig S1 edit X-axis title, it should be “Cycles” for PCR graph.” Your Re:”It should be the time. We did a real-time recording of the fluorescence signal intensity for both q-LAMP and q-PCR reactions using BIO-RAD CFX96.”

ARE YOU SURE?

Some places need improvement. 

Round 3

Reviewer 3 Report

I suggest to use Time (min) for all LAMP graphs.

For   Re:Thanks for the kind reminder, we have fixed that in line 14. 

Use “primer set… was”

For   Re: It should be the cycle threshold (Ct) values. The figures represented the real-time recording of the fluorescence signal intensity for both q-LAMP and q-PCR reactions, however the time was quantitated using the cycle threshold (Ct) values detected by Bio-Rad CFX96. Thus, the unit is cycle number not min and the X-axis should be Ct. We are really thanks for figure this mistake out, we now have fixed it.

 Fig 6 should have Time (min) on X-axis and no template control sample included.

 Re:Thanks for the kind reminder, we have fixed that.

You fixed the PCR graph but not LAMP graphs. In version 3, you switched to Cycles (Ct) on the X axis for all reaction graphs. If you want to report Ct instead of Cq provided by the machine it is OK. However, for graphs with isothermal LAMP reactions multiple manuscripts use Time (min) on the X axis (because cycle length can vary from 30 sec to 1 min) to make it clear. For graphs with PCR reactions, “Cycles” are always shown on the X axis (because they are described in the run method).  

For example, see  Paul, R., Ostermann, E., Chen, Y., Saville, A. C., Yang, Y., Gu, Z., Whitfield, A. E., Ristaino, J. B., and Wei, Q. 2021. Integrated microneedle-smartphone nucleic acid amplification platform for in-field diagnosis of plant diseases. Biosensors and Bioelectronics 187:113312.

For Re: Our previous determination of optimal conditions and specificity were all conducted by using mycelium samples, and this assay is also effective for mycelium samples. We now updated this information in the manuscript in line 333-336.

I don’t see any changes.

Round 4

Reviewer 3 Report

I appreciate your patience while adressing my comments. I just have one thing to mention: time can be measured in seconds, minutes, hours, or days. Please, fix the X-axis in your figures, use "Time (min)". Thank you.

Author Response

Thanks for your suggestions. We fixed the X-axis in your figures, use "Time (min)".